# Shielding Effectiveness Measurements of Drywall Panel Coated with Biochar Layers

**Patrizia Savi** [1,*] **, Giuseppe Ruscica** [2] **, Davide di Summa** [2] **and Isabella Natali Sora** [2]

[1] Department of Electronics and Telecom, Politecnico di Torino, C.so Duca degli Abruzzi 24, 10129 Torino, Italy
[2] Department of Engineering and Applied Sciences, Università di Bergamo, Dalmine, 24044 Bergamo, Italy;
giuseppe.ruscia@unibg.it (G.R.); disumma.d@gmail.com (D.d.S.); isabella.natalisora@unibg.it (I.N.S.)
*** Correspondence: patrizia.savi@polito.it**

**Abstract:** Shielding against electromagnetic interference (EMI) is a critical issue in civil applications generally solved with metal screens. In recent years, the properties of many composite materials filled with carbon nanotubes or graphene or materials with a carbon-based coating have been analysed with the aim of using them for electromagnetic shielding applications. Among other carbon materials, biochar, derived from biomass and characterized by high carbon content, emerges as a sustainable, renewable, environmentally friendly, and inexpensive material. In this paper, commercial biochar thermally treated at 750 °C is used to coat with several layers common building components such as drywall panel. Shielding effectiveness is measured in the frequency band 1–18 GHz for normal incidence and skew angles 10, 20 and 30 deg in a full anechoic chamber with double ridged, vertically and horizontally polarized broadband horn antennas. The results show that the proposed biochar-coated drywall panels provide a good shielding effectiveness compared to similar solutions, with the advantage of a less expensive and easier to realize building material.

**Keywords:** shielding effectiveness; biochar; eco-friendly material; drywall

## 1. Introduction

In recent years, the increasing exposure to several radio frequency (RF) sources (WiFi, Bluetooth® devices, mobile phones, etc.) has brought to attention the problem of mitigating the effect of these types of exposure to reduce the risks for human health [1]. Excessive exposure to such radiations increases the probability of tumors and other diseases.

Electromagnetic shielding in buildings highly depends on the electrical characteristics of the materials and on the thickness of the wall [2].

The shielding properties of materials are represented by shielding effectiveness (SE), which can be modelled with a transmission line model originally developed by Schelkunoff [3]. This model provides an exact solution only for an infinite homogeneous material sheet or layers of sheets for a plane wave with normal incidence. The decibel value of SE is given by the sum of three loss terms: absorption loss, reflection loss, and multiple reflections loss [4]. Carbon-based materials displaying EM shielding effects have been largely explored for both reflection and absorption mechanisms [5,6].

The shielding properties can also be quantified by measuring the shielding effectiveness (SE) of a flat material sample with a given thickness. The SE is the insertion loss expressed in decibels of the sample in free space with a normally incident plane wave.

In industrial applications, measurements of shielding effectiveness are generally performed by the IEEE standard (STD-299-2006) [7].

In modern buildings, the attenuation is generally low due to the reduced thickness of their envelope and the material used. In some cases, it is necessary to ensure that the living space is shielded from harmful radiation from telecommunication systems. Traditionally, metal sheets and wires (copper, nickel and stainless steel) were used for mitigating the

electromagnetic pollution in sensitive and confined environments. They exhibit excellent electrical conductivities and shielding effectiveness (SE) but show various drawbacks, such as high density, and are difficult to superimpose on a building envelope. The replacement of metallic materials with conductive polymer composites reduces the weight of the structures, permits obtaining more complex shapes, and imply simpler production processes [8]. The high cost of conductive fillers (e.g., carbon nanotubes, carbon fibers, and graphene nanosheets, etc.) and a certain low sustainability of polymers produced from hydrocarbon-based materials may limit their application as EMI shielding materials in the construction industry. Modern strategies to facilitate the use of EMI materials in non-structural building components are moving toward low-cost and sustainable materials such as biochar.

Among carbon-based materials, biochar (abbreviation for bio-charcoal) is a solid product from biomass pyrolysis [8–12], which commonly contains wood char, bamboo char, straw char and rice husk char. Biochar can also be obtained from sewage sludge [13–15]. Biochar has been used for various applications, such as the removal of pollutants from water [16] or as filler in composites, improving their mechanical and electrical properties [17]. Biochar has recently been gaining attention as a natural electromagnetic shielding product [18], which can be a promising resource that can be used on a building envelope [19–23].

Biochar differs from metals and other synthetic carbon-based materials because it shows shielding effectiveness mainly in the microwave range [24].

Recent studies showing the potential use of biochar as filler in matrices of interest for non-structural composites. The polymer-based composite (3 mm thick) with 80 wt% biochar exhibited a very high EMI SE of 48.7 dB (99.998% attenuation) at 1.5 GHz. The specific EMI SE was 39.0 dB cm$^3$/g, nearly four times higher than that of copper (10 dB cm$^3$/g) [25]. The biochar/(Fe,Ni)–ferrite composite can attain 46.36 dB at 0.49~1.43 GHz for 2 mm [26]. Values of SE > 20 dB at 8.2–12.4 GHz are achieved for the composites containing 18 wt% of wood biochar in cement matrix for 4 mm thickness [18]. Shielding effectiveness can also be computed from the knowledge of the complex permittivity of the material as in [18] for microwave frequencies up to 8 GHz.

The influence of ageing is a very important feature for biochar/cement composites: increasing wet curing increases the SE, while increasing ageing in air decreases the SE by 5 dB after 10 weeks. The effects of particle size, ageing, and water absorption of biochar particles [27] and durability [28] in cementitious composites have been recently analysed.

Protection against EM interference can also be obtained by using gypsium brick wall [24]. Gypsum is an electrical insulator and does not contribute to electromagnetic shielding. In the microwave region, the biochar/gypsum composites with 40 wt% of biochar show SEs of 13.6 ± 0.56 and 19.25 ± 1.8, at 5 and 6 GHz, respectively.

Another solution is to apply coating on wall surfaces or using drywall panels properly treated. The availability of drywall panels designed ad hoc for EMI shielding could guarantee great flexibility for the realization of protecting the environment for health care applications or to reduce the strong intensity of the fields in cases of a nearby wireless LAN.

There are several methods commonly used to measure shielding effectiveness (SE) of materials. Among the others, for samples of small dimensions, methods include the coaxial transmission line method [29–32] or waveguide methods (see, e.g., [33]) and the reverberation chamber method [34,35]. In the case of large size samples, free-space measurement [36], shielded room [7] and transmission measurements in anechoic chamber [37] are generally used.

In the coaxial method, the material under testing should fit in a precise enclosure while imposing stringent shape requirements. Reverberation chamber or free-space methods are generally suitable for materials covering large dimensions. When samples of large dimensions are required, carbon-based fillers such as graphene and carbon nanotubes are prohibitive due to their costs. Biochar that is very cheap and abundantly available is a good alternative.

In the transmission method, the signal transmitted through the sample is measured in an anechoic chamber by the help of two broadband antennas. The chamber is divided into

two areas by a metal wall approximating an infinite plane for which shielding effectiveness is defined. This method requires samples of large dimensions (200 × 200 mm or more); it is, therefore, particularly convenient in the case of low cost materials.

The attenuation of common building materials have been investigated for low frequency [2] and more recently at millimetre waves [38–41].

In this paper, the shielding effectiveness of drywall panels of dimensions 200 × 200 mm and thickness of 10 mm, coated with two layers and five layers of a paste based on a commercial biochar, is analyzed. These drywall panels were designed ad hoc at the University of Bergamo for EMI shielding. The transmission scattering parameter, $S_{21}$, is measured with two broadband horn antennas, vertically and horizontally polarized, in an anechoic chamber in the frequency band 1 GHz–18 GHz at normal incidence and for skew angles of 10, 20, and 30 deg. The results are compared with the literature data of the SE through gypsum/biochar composites, multiwalled carbon nanotube/cement composites, and biochar/cement composites.

In the case of properly treated drywall panels, the mechanical properties remain unchanged, and good shielding efficiency should be obtained not only for normal incidence but also for skew angles.

## 2. Materials and Methods

### 2.1. Materials and Samples Preparation

In this paper, a powder of commercial biochar obtained from wood biomass supplied by Carlo Erba is considered. The biochar contains 5 wt% of ashes, as reported in Section 3. The powders of hydroxyethyl cellulose (with 5 wt% moisture content at 20 °C) and 25% $NH_3$ (reagent grade) are used. The biochar is thermally treated at 750 °C for four hours. The biochar is then manually blended with water in the proportion 16:100 g. A 2% hydroxyethyl cellulose (8 g) and 25 drops of $NH_3$ are added and mixed for 1 min more until the mixture shows a viscous consistency. One side of commercial drywall panels (200 × 200 mm and 10 mm thick) is coated with the paste to form a layer with thickness less than 1 mm (see Figure 1). The panel is then cured at ambient conditions (T = 25 °C and relative humidity 50%) for 24 h. This procedure is repeated in order to deposit several layers. A 5-layer coating after drying at ambient conditions showed a final thickness of about 1 mm.

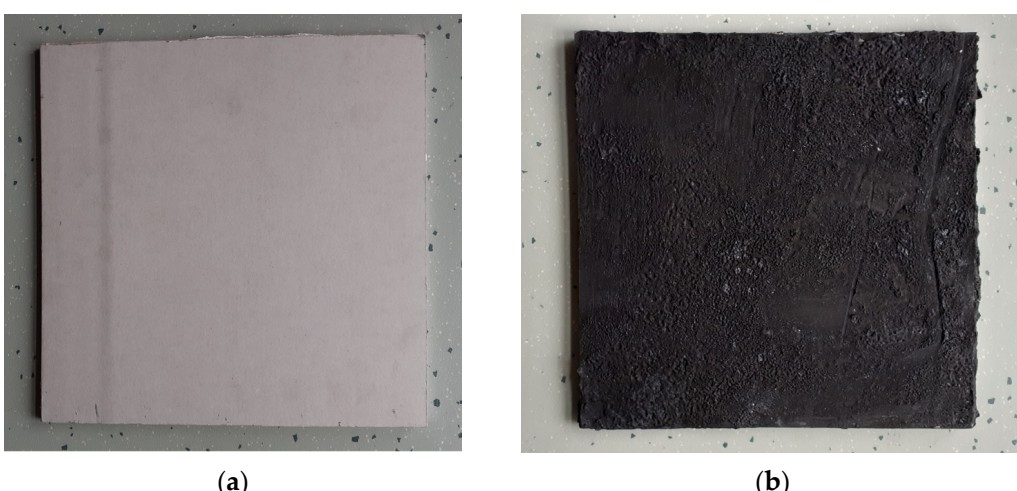

(**a**)                                                                                          (**b**)

**Figure 1.** (**a**) Drywall panel; (**b**) drywall panel coated with several layers of biochar paste.

### 2.2. Thermogravimetric Analysis

To figure out the thermal decomposition behavior of biochar, thermogravimetric (TGA) analyses are performed. The thermogravimetric measurement TGA of about 20 mg powder of biochar is carried out in air using heating from room temperature to 950 °C at slow heating rates (3 °C/min). The weight loss versus temperature curve shows several consecutive zones.

### 2.3. Shielding Effectiveness Definition and Measurement

To define shielding effectiveness (SE), a dielectric medium of thickness *d* in air and a plane wave incident from the left (see Figure 2) are considered. A portion of the incident wave is transmitted through the medium (dielectric slab). The SE of the dielectric slab is defined for the electric field, in decibels, as follows:

$$SE = 20 Log_{10} \frac{E_t}{E_i} \tag{1}$$

where $E_i$ is the electric field of the plane wave incident on the left, $E_t$ is the electric field transmitted after the dielectric slab. The same definition holds for the magnetic field.

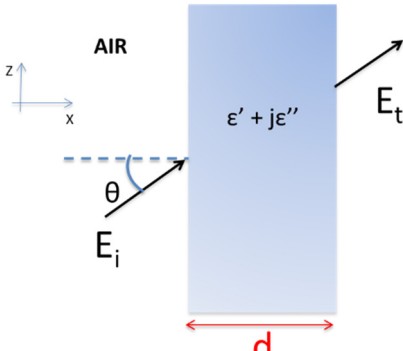

**Figure 2.** Shielding effectiveness of a dielectric slab to a uniform plane wave.

Note that the SE depends on the incidence angle (θ), on the thickness of the material (d), and on the complex permittivity of the material. Since the SE depends on the characteristic of the material, if the material is homogeneous, it could be evaluated from the knowledge of its complex permittivity [4,42]. In the case of samples with complex structure (multiple layers), the SE can be computed with a transmission line model [30] or determined based on measurements.

Transmission measurements are performed in a compact full anechoic chamber of dimensions 2 × 4 × 2 m with a metallic external enclosure at the Department of Electronic and Telecommunication (LACE Laboratory). Pyramidal microwave absorbers operating above 1200 MHz up to 90 GHz cover the walls, ceiling, and floor where possible. In the camera, 3-axis positioners are installed and measurements of gain, far-field patterns (amplitude and phase), and antenna-matching parameters can be performed. Measurements of the transmission are carried on with double-ridged, vertically polarized broadband horn antennas operating at 1–18 GHz and connected to a vector network analyzer (VNA, Agilent E8361A). Each material sample is placed at the center of the chamber. The samples are sufficiently light to be fixed to a thin wooden holder of dimensions 1.5 m × 1.5 m placed on the floor of the chamber between two wideband horns placed at the same height. The spacing between the opening of the two horns was 2.5 m. A sketch of the measurement setup is shown in Figure 3. The transmitting and receiving antennas as well as the sample under test are placed inside the chamber. The signal generator and receiver are placed outside the chamber in order to avoid interference. The sample are mounted on the window on a metallic wall that separates the anechoic chamber in two areas. SE is determined by comparing the signal levels measured with and without the sample between the two antennas. For each sample, the scattering parameter $S_{21}$ is measured as a function of frequency without the sample ($S_{21}$,ref) and with the sample made of drywall coated with biochar paste present ($S_{21}$,sample). The SE is then computed in dB as follows.

$$SE = 20 log_{10} \frac{\left|S_{21,sample}\right|}{\left|S_{21,ref}\right|} \tag{2}$$

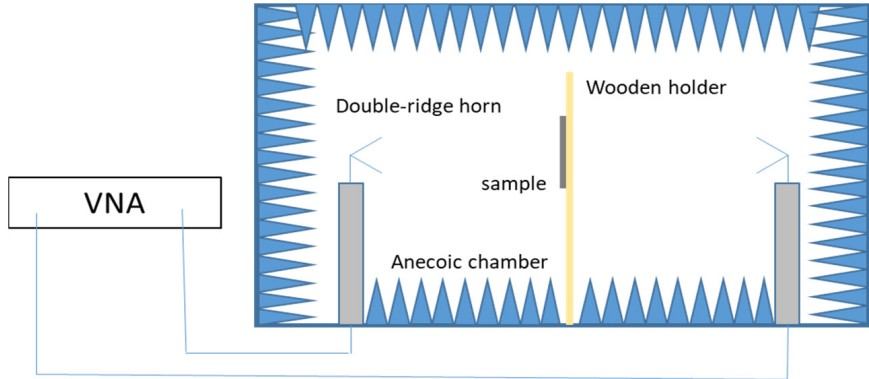

**Figure 3.** Sketch of the measurement system.

### 3. Results

TGA experiments are used to perform the analysis of biochar. In particular, the water and carbon content of the used biochar is investigated. In Figure 4, the TGA curve of biochar is reported. The first zone of rapid (temperatures below 100 °C) weight loss (weight loss up to 16%) is the step of the evaporation of the physically adsorbed water. Biochar combustion is completed at about 500 °C, mainly due to the combustion of the graphitic carbon fraction, which is about 74 wt% of the biochar. At the temperature reached at the end of the experiment (950 °C), a residue of around 5 wt% with respect to the initial amount is observed.

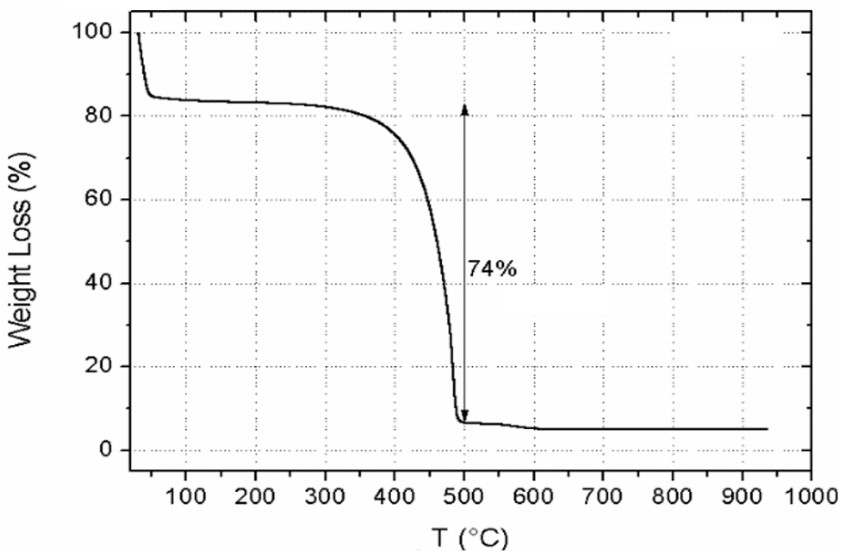

**Figure 4.** TGA curve of biochar.

The measurement setup used for the measurements of the SE is shown in Figure 5. A wooden holder covered with metal to simulate a metal wall is inserted between the two antennas. The holder has a central hole on which the sample is fixed and can be rotated to perform skewed incidence measurements. Results are shown in Figure 6 for the three samples coated with two layers of biochar and in Figure 7 for the three samples coated with five layers of biochar. In all plots, the black curve is the average value. The SE is around 10dB in the entire frequency band in the case of the two-layer samples for both vertical and horizontal polarization, whereas, in the case of five-layer samples, it is −18 dB at 1 GHz and decreases to 30 dB at 18 GHz. In Figures 8 and 9, sample 1 of the two-layer set and sample 3 of the five-layer set are analyzed varying the angle of incidence from 0 to 30 deg.

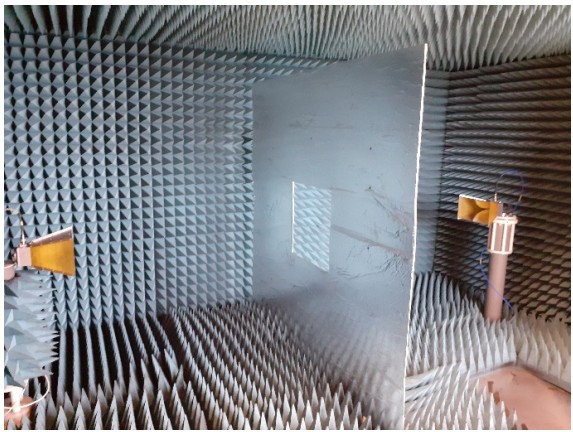

**Figure 5.** Measurement setup for transmission measurements of the drywall.

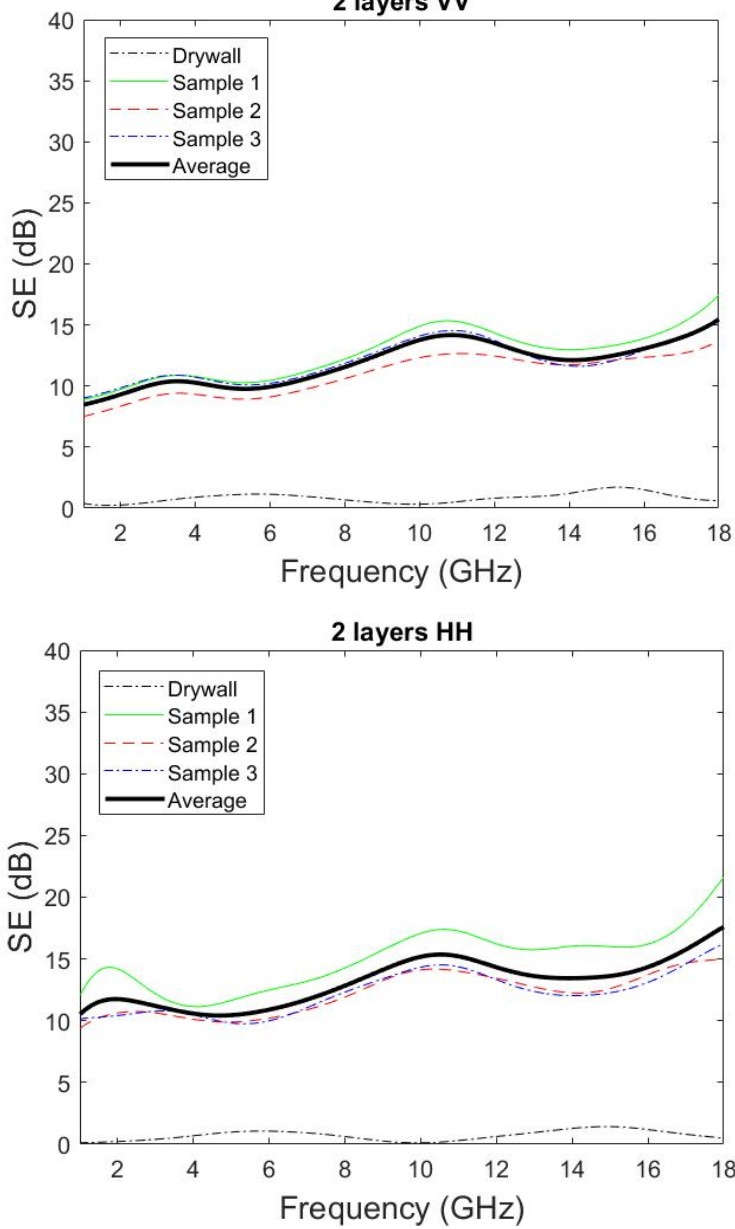

**Figure 6.** Plots of the SE versus frequency for 2-layer samples. Vertical polarization (**top panel**); horizontal polarization (**bottom panel**).

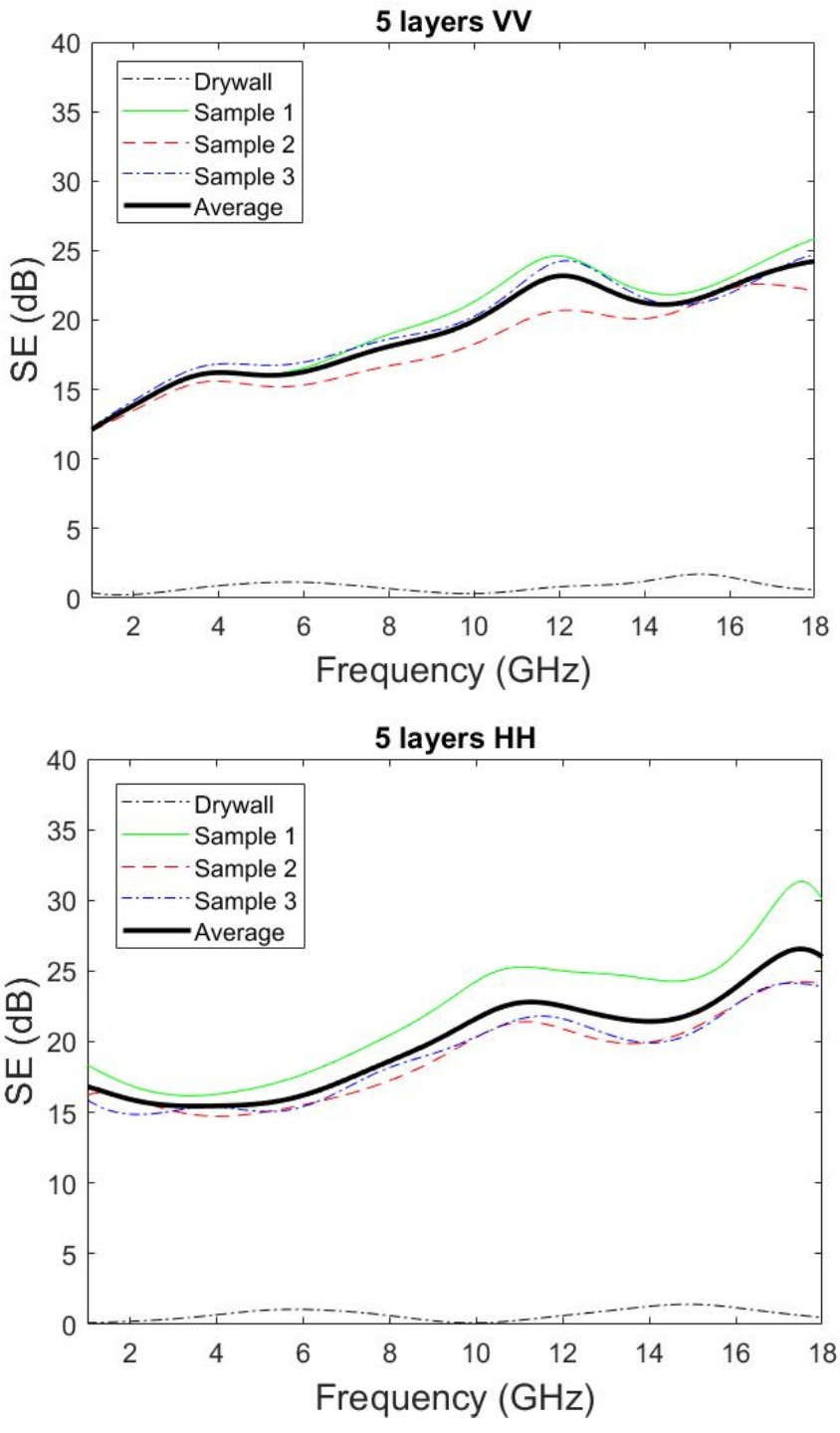

**Figure 7.** Plots of the SE versus frequency for 5-layer samples. Vertical polarization (**top panel**); horizontal polarization (**bottom panel**).

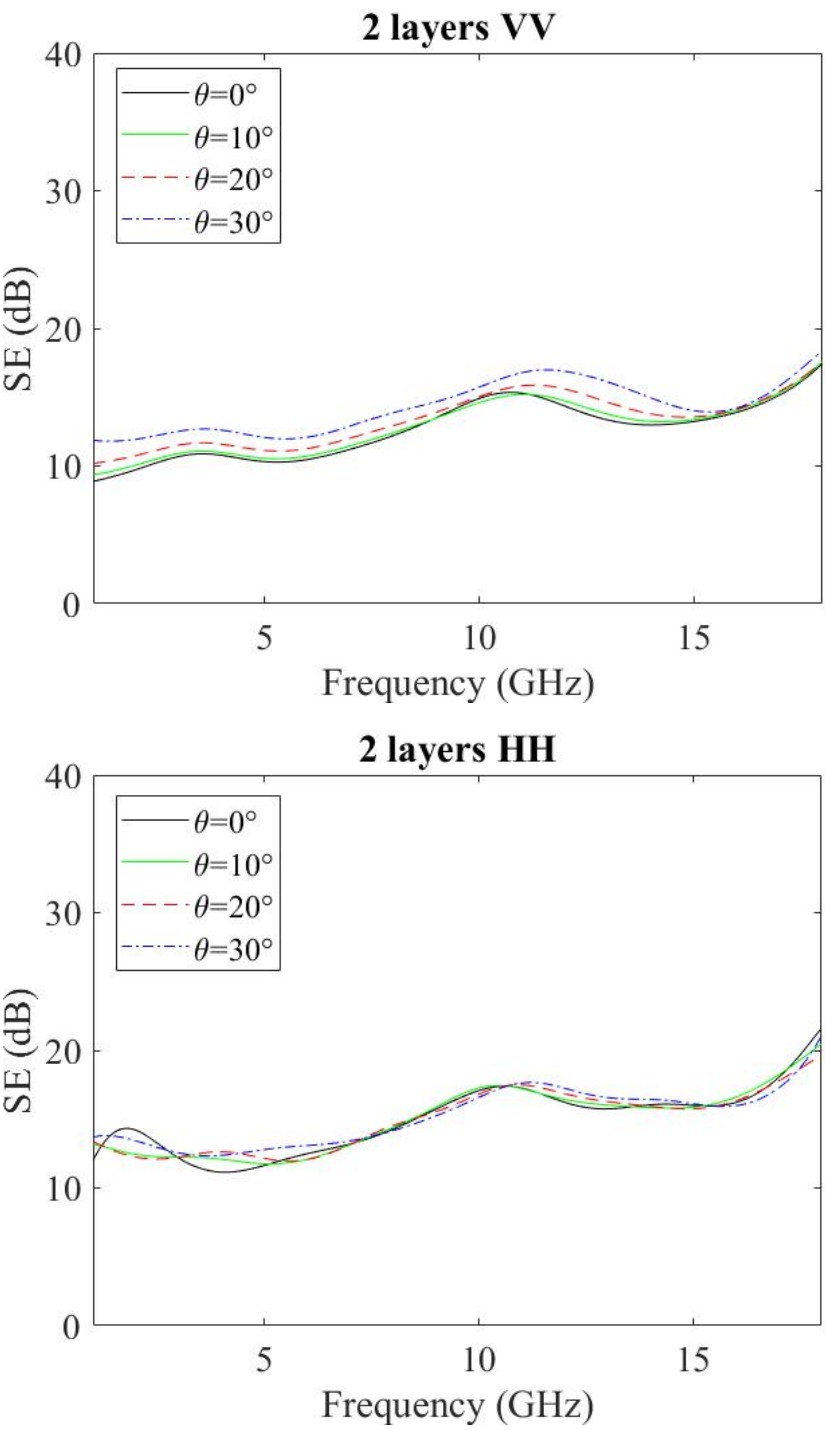

**Figure 8.** Plots of the SE versus frequency of a sample with 2-layer biochar coating for different angle of incidence. Vertical polarization (**top panel**). Horizontal polarization (**bottom panel**).

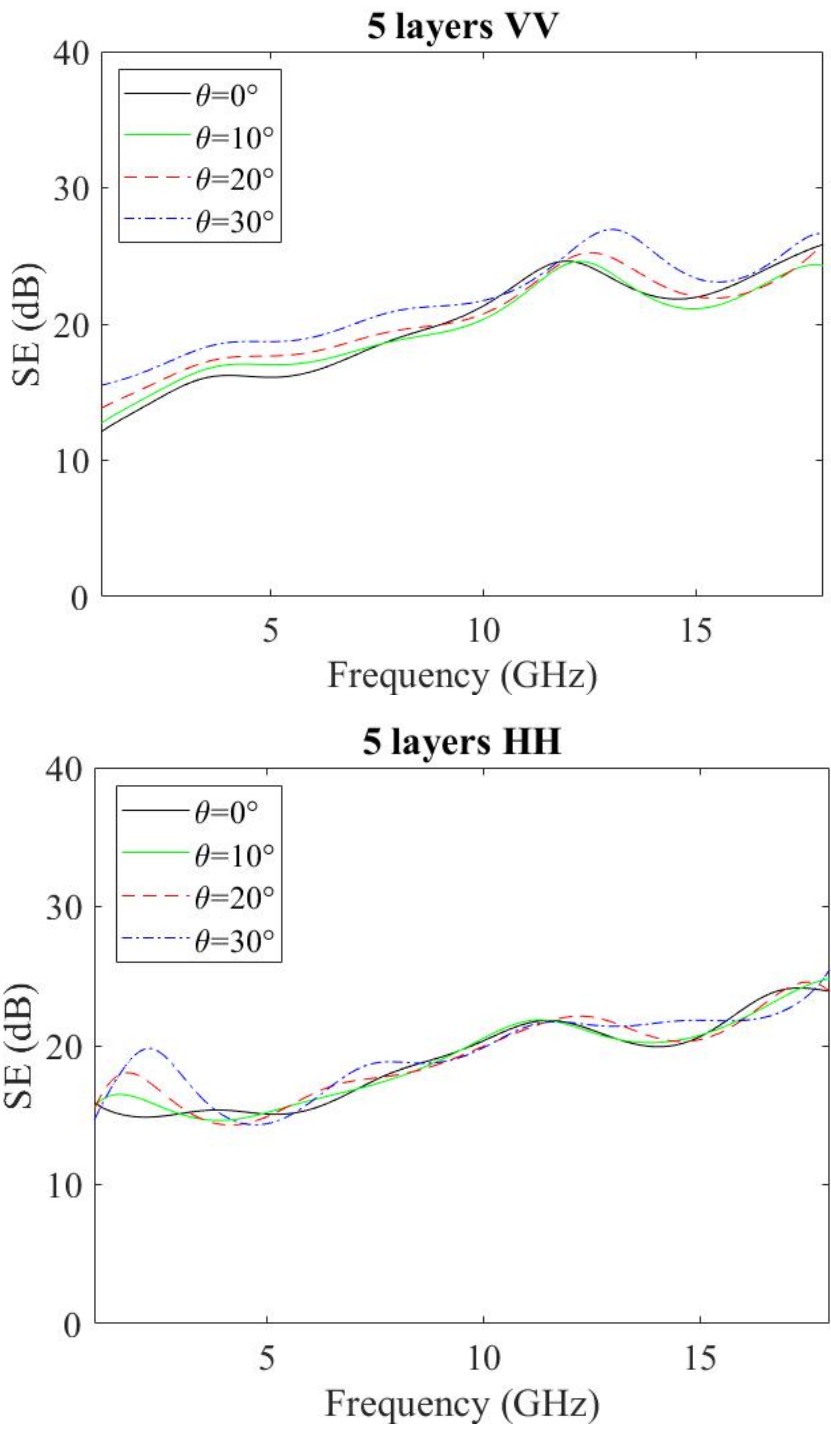

**Figure 9.** Plots of the SE versus frequency of a sample with 5-layer biochar coating for different angles of incidence. Vertical polarization (**top panel**). Horizontal polarization (**bottom panel**).

## 4. Discussion

The SE depends both on the thickness of the sample and of the electrical characteristics of the materials. Therefore, a rigorous comparison with already published data is very difficult.

To have an indication of the possible values of SE for various materials suitable for the construction sector, Table 1 compares the SE at 2 GHz and in the range 8–12 GHz of five-layer biochar drywall of this work with other configurations even if the thicknesses of the samples and materials are quite different. According to Table 1, the drywall panels

coated with biochar, proposed in this work, provide a greater SE behavior than the other materials at 2 GHz with the exception of 80% *w/w* biochar/ultra-high molecular weight polyethylene (UHMWPE)/linear low-density polyethylene (LLDPE) composite [25], which, moreover, is 3 mm thick. Considering the 8–12 GHz frequency range, drywalls coated with biochar have a slightly worse performance than multi-walled carbon nanotube/cement composites [43,44] with 30% *v/v* biochar/cement [1,19]. However, it is worth noting that the materials proposed in this work present a lower cost and an easier preparation procedure. This last consideration is very important when considering an industrial process for mass production.

**Table 1.** Shielding effectiveness data for some biochar-based composites or drywalls for EMI shielding applications.

| Materials | Thickness (mm) | SE (dB) 2 GHz | SE (dB) 8–12 GHz | Ref |
|---|---|---|---|---|
| Gypsum based drywall coated with biochar | 10 mm drywall panel + 1 mm 5-layer biochar | 18 | 21 | This work |
| Gypsum−40% *w/w* biochar composites with 2 cardboard | 2.5 mm cardboard + 10 mm composite + 2.5 mm cardboard | 10 | 20 | [24] |
| Composites, biochar 0.5% *w/w* of cement | 10 mm | 6 | 9 | [43] |
| Composites, biochar 18% *w/w* of cement | 4 mm | n.a. | 19–23 | [21] |
| Composites, 30% *v/v* biochar/cement | 30 mm | 9 | 25 (@ 8 GHz) | [19] |
| Composites, 15% *w/w* multi-walled carbon nanotube/cement | 2 mm | n.a. | 27 | [44] |
| Composites, 80% *w/w* biochar/ultra-high molecular weight polyethylene (UHMWPE)/linear low density polyethylene (LLDPE) | 3 mm | 49 (@ 1.5 GHz) | n.a. | [25] |

## 5. Conclusions

In this work, we report the measured SE of drywall panels with two-layer or five-layer biochar coating applied onto one face. The coating contains commercial biochar from wood biomass.

The SE of panels covered with two-layer and five-layer biochar coating was measured with an insertion loss method in an anechoic chamber with double-ridged, vertically polarized broadband horn antennas operating at 1–18 GHz connected to a vector network analyzer (VNA, Agilent E8361A).

From the obtained results, the following conclusions can be drawn for normal incidence:

- SE around 10 dB is obtained in the frequency band 1–18 GHz for two-layer panels for both vertical and horizontal polarization, whereas the SE of the drywall panel is almost 0 dB as expected.
- The SE for five-layer panels is greater than in the case of two-layer and it was found to be 17 dB at 1 GHz and 25 dB at 18 GHz.
- There is a good reproducibility of the measurements for the various samples.
- Then, one panel with two layers and one panel with five layers were considered and measured for skew incidence from 0 deg to 30 deg:
- In the case of the two-layer coating, the results are not very influenced by changing the incidence angle.
- In the case of the five-layer coating, results are slightly different for different angles of incidence.

The promising potentialities of drywall panels coated with a biochar paste to improve shielding to EM interference in buildings is confirmed for the normal incidence for both vertical and horizontal polarization. These results are confirmed also for skew incidence up to 30 deg. Drywall panels coated with several layers of biochar paste are low cost and easy to fabricate, and they can offer great flexibility to realize a protected environment

for health care applications (chemotherapy equipment and tomography), reducing the strong intensity of the electromagnetic fields in the case of nearby electronic equipment or telecommunication repeater.

**Author Contributions:** Conceptualization, P.S., I.N.S. and G.R.; methodology, P.S., I.N.S. and G.R.; validation, P.S.; formal analysis, P.S.; investigation, D.d.S.; resources, P.S., I.N.S. and G.R.; data curation, P.S.; writing—original draft preparation, P.S., I.N.S. and G.R.; writing—review and editing, P.S., I.N.S. and G.R.; visualization, P.S.; supervision, P.S., I.N.S. and G.R. All authors have read and agreed to the published version of the manuscript.

**Funding:** This research received no external funding.

**Acknowledgments:** The authors would like to thank Renato Pelosato for TGA measurements and Gianluca Dassano for the shielding effectiveness measurements.

**Conflicts of Interest:** The authors declare no conflict of interest.

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
