# Peer review of "Shielding Effectiveness Measurements of Drywall Panel Coated with Biochar Layers"

_electronics, doi:10.3390/electronics11152312_

Round 1
Reviewer 1 Report
The paper presents essentially the measurement results of the shielding effectiveness (SE) of a biochar coating. Description of the biochar material and of the preparation of the samples are given, with some information on thermogravimetric analysis. The SE is measured with a usual “insertion loss” technique.
Overall, as the paper is limited to experiments only, probably journals more focused on materials should be considered for publication.
As regards the SE measurement results, I have the following comments.
First, although the specific measurement procedure adopted in the work is not standardized, the authors should indicate some standard/report used as a reference for this measurement (e.g., IEEE STD 299).
Moreover, it would be interesting to know what is the dynamic range of the test setup.
Are the SE measurements for horizontal or vertical polarisation of the antennas?
The SE values shown are very smooth: have they been averaged somehow?
Concerning Fig. 7, why the SE of the 5 layer shield is influenced more by different incidence angles than the SE of the 2-layer shield? is there an explanation for this?
The first paragraph of Sec. 4 (the Discussion) seems to contain general information (there is a mention to the coaxial waveguide SE measurement, which is not carried out in this work), and therefore it could be moved earlier in the text.
Author Response
please see the file in attachment

Reviewer 2 Report
The subject of this paper is interesting. However, needs minor revision.
1. Include more points related to the importance of Biochar in EMI shielding applications using the very recent references.
2. Include the recently reported results of Biochar in EMI shielding applications.
3. Add the purity of the used chemicals.
4. The authors should include the DTA graph of Biochar.
5. Figures 6 & 7: Graphs should be in different color; reformat the figures 6 & 7 (a,b).
6. What is the conclusion from table 1? If possible include more recent reports and compared them with present results.
7. Rewrite the conclusion part.
8. Check the reference format: Ref. 04, 08, 15, 18, 22.
Author Response
pleae see the file in attachment
Author Response
please see the file in attachment

Round 2
Reviewer 1 Report
The paper has been improved following the suggestions. The paper remains essentially a presentation of experimental results of the shielding effectiveness of biochar coated drywall panels. Thus, it would have been interesting to see the results for the horizontal polarisation of the antennas, or for a different dimension of the material sample. Moreover, if the 5-layer results may be affected by irregularities in the coating, measurement on other 5-layer samples could have been useful for comparison and confirmation of this conclusion.
The measurement results show that some SE can be achieved at frequencies above 10 GHz; potential applications of this shielding technique should thus be indicated.
The first part of the conclusions actually represents a summary of the paper.
Reviewer 3 Report
aqccept
Author Response
This reviewer did not write any further comments